# Stuck in the MOS pit:
# A critical analysis of MOS test methodology in TTS evaluation

*Ambika Kirkland, Shivam Mehta, Harm Lameris, Gustav Eje Henter, Éva Székely, Joakim Gustafson*

Division of Speech, Music & Hearing, KTH Royal Institute of Technology, Stockholm, Sweden

`kirkland@kth.se, smehta@kth.se, lameris@kth.se, ghe@kth.se, szekely@kth.se, jkgu@kth.se`

## Abstract

The Mean Opinion Score (MOS) is a prevalent metric in TTS evaluation. Although standards for collecting and reporting MOS exist, researchers seem to use the term inconsistently, and underreport the details of their testing methodologies. A survey of Interspeech and SSW papers from 2021-2022 shows that most authors do not report scale labels, increments, or instructions to participants, and those who do diverge in terms of their implementation. It is also unclear in many cases whether listeners were asked to rate naturalness, or overall quality. MOS obtained for natural speech using different testing methodologies vary in the surveyed papers: specifically, quality MOS is on average higher than naturalness MOS. We carried out several listening tests using the same stimuli but with differences in the scale increment and instructions about what participants should rate, and found that both of these variables affected MOS for some systems.

**Index Terms**: speech synthesis, TTS evaluation, mean opinion score, text-to-speech, neural TTS

## 1. Introduction

Text-to-speech systems are designed for many different purposes and contexts, but typically researchers developing such systems wish to subjectively evaluate their overall performance in some way. Though the suitability of these measures for evaluating synthesized speech has been questioned [1, 2, 3], the Mean Opinion Score (MOS) remains a ubiquitous means of evaluating TTS. Given its widespread use, one would assume that when researchers talk about MOS we are all talking about the same thing. The existence of an ITU standard for measuring audio quality [4] may reinforce this assumption Although the standard does state that MOS typically involves an absolute category rating on a 5-point scale (1-Bad, 2-Poor, 3-Fair, 4-Good, 5-Excellent) the ITU standard for MOS reporting [5] emphasizes that there are many potential deviations from this setup and states that reporting details such as "whether [the] rating scale is discrete or continuous", the rating scale labels, and which instructions and questions the participants receive is *mandatory*. Hence, per the standards themselves, merely referencing the standards does not ensure that everyone is on the same page. We cannot be certain about which methodological choices researchers make when they carry out listening tests unless they describe those choices in detail.

Unfortunately, these details are often missing. In a survey of Interspeech papers from 2014, [6] found that many authors failed to mention crucial details of the evaluation such as the number of listeners, listener demographics, and what kind of stimuli were used in the experiment. A similar analysis of Interspeech 2022 papers carried out by [7] showed that there are still deficiencies in reporting instructions to participants, evaluator demographics, and how much participants are paid.

Extending the work of [6], we surveyed a number of TTS papers which use MOS to evaluate text-to-speech systems, with a focus on the methodology of the rating task. In order to limit this survey to a manageable scope we focused on Interspeech papers from 2021 and 2022 and Speech Synthesis Workshop papers from 2021, all accessed via the archive of the International Speech Communication Association (ISCA) [1]. While this covers only a fraction of TTS research, we believe it represents a reasonable snapshot of recent work and can provide useful insights to TTS researchers.

Beyond getting a clearer picture of how TTS is evaluated, we also want to understand whether differences in methodology matter. Could changing details of how listening tests are carried out change the results? Using a survey of recent TTS papers as a starting point, we explore 1) which differences in MOS test implementation are most common and 2) how, if at all, these differences might affect MOS. Based on our findings we suggest a few concrete steps that could be taken to improve transparency and consistency in subjective evaluation of TTS.

## 2. Survey of MOS methodology

### 2.1. Selection of papers

We surveyed a total of 133 papers from Interspeech 2021, SSW 2021, and Interspeech 2022, which involved speech synthesis and used a subjective listening test in their evaluation which they specifically referred to as a MOS test. Papers which *only* used similarity MOS or CMOS were not included, nor were those using only a 100-point MUSHRA test. However, 5-point "MUSHRA-like" MOS tests were included so long as these were referred to as MOS tests by the authors. In total 77 papers were included from Interspeech 2022, 48 from Interspeech 2021 and 8 from SSW 2021.

### 2.2. Variations in MOS listening test design

#### 2.2.1. What does MOS measure, anyway?

One possible variation in MOS test design is which aspects of the speech samples participants are asked to evaluate: typically, either naturalness or overall quality. Determining which of these was measured in the surveyed papers was not trivial. First, as seen in Table 1 a substantial number of papers (16.5%) are not clear about whether their MOS values reflect naturalness or quality, stating that they conducted subjective tests or asked participants to rate the stimuli on a 5-point scale without providing additional detail.

---

[1] https://www.isca-speech.org/archive/index.html

Table 1: *What MOS is stated to measure, as reported in 133 TTS papers at Interspeech and SSW*

| Measure | Count |
|---|---|
| Naturalness | 67 (50.4%) |
| Quality | 30 (22.6 %) |
| Unknown/unclear | 22 (16.5%) |
| Multiple | 9 (6.8 %) |
| Other | 5 (3.6%) |

Even in cases where researchers stated that their MOS specifically represented either quality or naturalness, it is usually not clear whether this reflects the actual instructions to participants, or merely the researchers' own ideas about what MOS is *intended* to measure. Only 2 papers provided the actual question posed to participants, and only 31% included at least a partial description of the scale labels. When scale labels include the word "natural", it is reasonably clear that participants knew they were meant to rate naturalness. In cases where a "naturalness" scale was labelled from "Bad" to "Excellent", however (or when no labels were specified), we do not know whether researchers simply asked participants to rate speech from 1 to 5 and assumed the responses reflected naturalness, or whether they explicitly asked them to rate naturalness. However, we have mostly taken the language used by researchers at face value. The evaluations categorized as measuring quality or naturalness in Table 1 reflect what the authors stated they wanted to measure.

On the other hand, cases where the authors seemed to fully conflate quality with naturalness and referred to them interchangeably without clarifying which they asked participants to rate are included in the "unknown/unclear" category. The "other" category includes papers which used MOS to measure something other than naturalness or quality, while the "multiple" category includes papers which stated that participants rated quality and/or naturalness in addition to some other aspect of the speech (e.g., "naturalness of voice and appropriateness of pronunciation for the particular language or dialect").

### 2.2.2. MOS scale increments

Another detail which is underspecified in the surveyed papers is the number of increments on the rating scale and whether it is discrete or continuous. As shown in Table 2, more than 75% of papers did not state which increments they used. Those which did specify are fairly evenly divided between full-point and half-point increments.

Table 2: *MOS scale increments reported in TTS papers at Interspeech and SSW (2021-2022)*

| Measure | Count |
|---|---|
| Unknown | 100 (75.2%) |
| Half-point | 15 (11.3%) |
| Full point | 18 (13.5%) |

### 2.2.3. MOS scale labels

As already noted, the majority of researchers did not specify how they labelled their MOS scale. Among those who did

Table 3: *Examples of MOS scale labels reported in 2021-2022 Interspeech and SSW papers. When no intermediate labels are given, only the endpoints of the scale were specified in the paper.*

| Scale labels | Count |
|---|---|
| No labels specified | 92 (69%) |
| 1 (Bad), 2 (Poor), 3 (Fair), 2 (Good), 5 (Excellent) | 18 (13.5%) |
| Bad to Excellent | 4 (3%) |
| Very unnatural to Very natural | 4 (3%) |
| Very bad to Very good | 2 (1.5%) |
| Completely unnatural to Completely natural | 2 (1.5%) |
| Very bad to Excellent | 2 (1.5%) |
| Naturalness: 1 (Very annoying) 2 (Annoying) 3 (Slightly annoying) 4 (Perceptible but not annoying) 5 (Almost real) | 1 (<1%) |
| Excellent to Worse | 1 (<1%) |
| Negative to Positive | 1 (<1%) |
| 1 (Poor), 2 (Bad), 3 (Fair), 2 (Good), 5 (Excellent) | 1 (<1%) |
| 1 (Very unnatural), 2 (Rather unnatural), 3 (Neither), 4 (Rather natural) 5 (Very natural) | 1 (<1%) |
| Non-intelligible to Excellent naturalness | 1 (<1%) |
| Unintelligible to Excellent quality | 1 (<1%) |
| [Unspecified] to Completely natural speech | 1 (<1%) |

specify, most used a scale with the labels 1 (Bad), 2 (Poor), 3 (Fair), 2 (Good) and 1 (Excellent), as recommended by the ITU standard [4]. This type of scale was used for both naturalness and quality MOS. However, other variations are shown in Table 3. In some cases the labels wildly diverge from the typical labelling scheme (e.g., rating naturalness from "very annoying" to "almost real"). In others they differ slightly in ways that alter the distance or order of categories, such as starting with "very bad" instead of "bad" (which shifts the low end of the scale lower), switching "bad" with "poor" (which disrupts the continuum of categories from worst to best), or using labels on one end of the scale that map to different measures than the other end (e.g., asking listeners to rate from "non-intelligible" to "excellent naturalness" even though "intelligible" and "natural" are not antonyms).

### 2.2.4. References to ITU standards

The ITU standards for measuring and reporting MOS stress that there are degrees of freedom in setting up listening tests, and properly reporting evaluation methods and listener characteristics is part of complying with the standards [4, 5]. Still, these standards do lay out some basic assumptions and provide examples of how MOS tests could be carried out [8]. If authors state that they have followed a standard, it might be fair to assume that they have used the default parameters outlined there. However, only one of the surveyed papers referenced an ITU standard relating to MOS when describing their evaluation, and this paper deviated from the quality scale described in the source they cited [4] by asking participants to rate naturalness instead of quality.

## 3. Effects of test design on MOS

Though it is clear that there is substantial underspecification and variation in terms of how MOS is used, do any of these variations really matter? Are the many small choices every researcher makes when designing an evaluation mere stylistic differences, or can they meaningfully influence the outcome?

### 3.1. Previous work

A relatively small number of studies have investigated whether variations in test design can affect MOS or similar rating scales. For example, [9] found a strong linear relationship between ratings of video quality on continuous and discrete scales with different numbers of points (5, 9 or 11), but the highest quality video received lower ratings on the 9-point scale. There is also evidence from [10] that category labels can impact ratings of audio quality. Despite a high correlation between ratings of audio samples on a quality scale (with 5 labels ranging from "Bad" to "Excellent"), an impairment scale (with 5 labels ranging from "Very annoying" to "Imperceptible") and an unlabelled scale, the unlabelled scale resulted in overall higher ratings than the impairment scale while the worst-scoring audio got higher average ratings on the quality scale than on the impairment scale.

These studies suggest a mostly linear relationship between quality scores on different types of scales, but showed that differences in the number of points on the scale or the labels used may result in overall higher or lower ratings, with particularly high- or low-quality stimuli being especially sensitive to such differences. However, neither is directly comparable to MOS evaluations of TTS voices. In both studies, multiple features of the scales varied between conditions, making it hard to isolate which may have been responsible for differences in ratings. But perhaps more importantly, the stimuli consisted of music recordings or videos with relatively coarse distortions and are hence quite unlike the stimuli used in TTS evaluation.

More directly applicable is evidence from [11] showing that small changes in the wording of instructions can influence naturalness ratings of spoken utterances. When rating clips of spontaneous utterances and versions of these utterances read by the same speakers, listeners rated the spontaneous utterances higher in terms of naturalness than the read utterances. However, this held true only if they were asked to rate naturalness generally, or were asked to rate naturalness as if they had heard the utterances in a conversation. When asked to rate how natural the utterances would sound if someone were reading them aloud, there was no preference for either type of utterance. Although this study involved natural speech rather than TTS, it suggests that these kinds of details could conceivably impact MOS. It has also been shown by [12] that participants treat the task of rating the naturalness of synthesized speech in context differently than rating contextual appropriateness, and they found differences in MOS between these two tasks with very small changes to the instructions. Still, there is not much work that systematically investigates how common variations in methodology like those we encountered in our survey might influence evaluations of synthesized speech.

### 3.2. Analysis of previous TTS evaluations

One way of assessing the impact of test design is to compare previously reported MOS from experiments using different methods, for example, comparing MOS obtained using a half-point scale to those obtained with a full-point increment scale. A limitation of this approach is the amount of variation in the stimuli, type and number of systems, languages, listener demographics, etc., used across different listening tests, which could either wash out differences, or introduce bias if multiple aspects of test design systematically co-vary. Nonetheless, if we look only at ratings of non-vocoded ground-truth recordings, i.e., natural speech, it may be possible to form some tentative insights that could lead to testable hypotheses about the impact of listening test methodologies. To that end, we analyzed a sub-

set of the surveyed Interspeech and SSW papers to determine whether there was a relationship between testing methodology and the ratings of natural speech reported in these papers.

To compare ratings from quality scales to those from naturalness scales, we selected papers which clearly stated which of these they intended to measure. A paper which switched the lowest two labels on the most commonly used "Bad" to "Excellent" rating scale (using 1 to indicate "Poor" and 2 for "Bad") was excluded. We also excluded ground-truth (GT) audio that was vocoded or manipulated in any way (other than loudness normalization). Ideally, we would have also included rating scale increments as a variable, but the scale increment was specified in only a handful of papers that met our criteria.

MOS of GT audio were taken from the results reported in the papers. In cases where multiple experiments were conducted, we used the scores from each of these experiments as separate data points. The analysis included 75 ground-truth mean opinion scores from 50 papers. The mean GT MOS was higher in experiments reporting quality MOS ($4.46 \pm 0.11$) than in those reporting naturalness MOS ($4.25 \pm 0.08$). An independent samples t-test (two-tailed) showed that this difference was significant, $t(73) = 3.24$, $p < 0.005$.

Comparing scales with different increments was complicated by the small amount of data. Of the 33 papers reporting scale increments, only 14 included GT MOS, for a total of 19 data points. The mean GT MOS for half-point scales ($4.34 \pm 0.14$) was higher than that reported for full-point scales ($4.14 \pm 0.37$), but this difference was not significant, $t(17) = 1.31$, $p = 0.21$. However, given the small number of observations and the fact that we lacked the data to tease apart the effect of scale increments from that of the measure used, or other differences in methodology, these results are not very conclusive. Furthermore, we cannot know in most cases whether participants were actually asked to rate quality or naturalness. Hence, we followed up with an experimental evaluation of the effect of scale increments and rating task instructions.

## 4. Experimental evaluation of MOS design

In order to assess the effect of variations in listening test methodology in a more controlled fashion, we designed 4 different versions of a MOS test which varied in terms of the scale increments and whether participants were asked to rate the naturalness or overall quality of the stimuli, while holding constant as many other factors as possible.

In this experiment, you will listen to a number of speech samples and **rate the overall quality** on a scale from 1 (Bad) to 5 (Excellent).

In this experiment, you will listen to a number of speech samples and **rate how natural they sound** on a scale from 1 (Bad) to 5 (Excellent).

Figure 1: *Variations of the listening task instructions. The text shown in bold font was bold on the instruction screen as well.*

The first variable in our evaluation was which aspect of the speech participants were asked to rate. The wording of these different versions varied in two places: on the task instruction page, and in a text box above the rating scale reminding participants what they should evaluate. The variations in task instructions are shown in Figure 1. The text displayed above the rating

scale read "Listen to the speech samples and rate their overall quality" in the quality version, and "Listen to the speech samples and rate how natural they sound" in the naturalness version.

We also varied the number of rating scale increments. Figure 2 shows the half-point and full-point versions of the rating scale. The scales were otherwise identical, with the same labels ("Bad" to "Excellent"). Both scales were discrete, meaning that it was not possible to select a value between the ticks on the rating scale (i.e., ratings of 1, 2, 3, 4 and 5 could be selected on the one-point scale and ratings of 1, 1.5, 2, 2.5, 3, 3.5, 4, 4.5 and 5 were allowed on the half-point scale).

These two variables were crossed for a total of four conditions: a naturalness MOS with a full-point scale, a naturalness MOS with a half-point scale, a quality MOS with a full-point scale, and a quality MOS with a half-point scale. Each participant completed only one of these versions of the listening task.

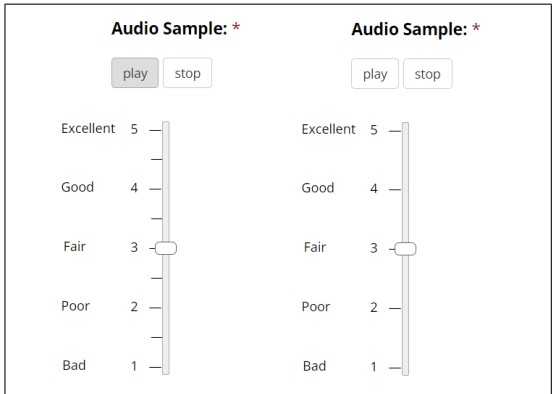

Figure 2: *Rating scales with half- and full-point increments*

### 4.1. TTS systems and stimuli

Because the focus of this paper is on methodology and not the performance of any particular TTS system, we chose systems which cover a range of different architectures and used official, publicly available implementations of these systems. All models were pre-trained on the same corpus (LJSpeech[2] which is a single speaker read speech corpus from a female speaker). We used the official checkpoints provided in the repositories and called the inference script of the original source code. We also used a universal checkpoint for the HiFi-GAN neural vocoder [13] used to reconstruct the waveforms, which was not fine-tuned on any particular TTS model.

The systems used to synthesize stimuli for the listening tests were Tacotron 2[3] [14] (referred to as T2), Glow-TTS[4] [15] (GTTS) and VITS[5] [16]. T2 is an autoregressive neural TTS system which uses attention to align text and audio. GTTS, on the other hand, is non-autoregressive and uses a monotonic alignment search instead of attention. Like T2, it is an acoustic model, representing audio with mel spectrograms. Finally, VITS is an end-to-end architecture generating waveforms directly from input text, without the need for a vocoder. Like GTTS, it is non-autoregressive and uses a monotonic alignment search, but integrates a neural vocoder (similar to the generator of HiFi-GAN) with an acoustic model similar to that of GTTS.

---

[2]https://keithito.com/LJ-Speech-Dataset/
[3]https://github.com/NVIDIA/tacotron2
[4]https://github.com/jaywalnut310/glow-tts
[5]https://github.com/jaywalnut310/vits

Because the subset of papers we analyzed in section 3.2 use ground-truth audio, as is also the case in evaluations in TTS challenges such as the Blizzard Challenge [17, 18, 19], we also included non-vocoded natural speech from the LJSpeech corpus in the evaluations. The ground-truth audio and corresponding text synthesized by the 3 TTS systems described above were 40 utterances selected at random from the test set of held-out LJSpeech utterances listed in the Tacotron 2 repository. Note that we cannot be certain that these utterances were also held out from the training of the other two systems.

### 4.2. Listening tests

Participants were recruited via Prolific and completed the listening test on the presentation platform Cognition. The participants were self-reported native speakers of English from the UK, US, Australia, Canada, New Zealand, and Ireland. Their ages ranged from 22-71, with a mean age of 37.6. 52% of participants identified as male, and 47% as female.

Participants were asked to confirm that they were wearing headphones and had no hearing impairments. A browser check insured compliance with the instructions to use a desktop browser compatible with the experiment. Furthermore, we used attention checks at random points: at the very end of two of the audio clips, listeners heard an instruction to choose a specific number on the rating scale. Data from participants who failed any attention checks were excluded from the statistical analysis. In total, 104 participants passed the screening questions and attention checks.

Participants were evenly divided between the four versions of the listening test (26 participants each). Each participant listened to 4 different versions (the ground-truth audio and three synthesized versions) of 12 utterances randomly selected from the total set of 40 utterances, meaning that each participant rated 48 samples (excluding the attention checks) and each system was rated by all 104 participants (a total of 1,248 ratings per system). The stimuli were presented in random order. The median completion time for the task was 10 minutes and 17 seconds.

### 4.3. Results

To determine whether scale increments or instructions about what participants should rate affected mean opinion scores, we carried out a 2 (measure: naturalness, quality) × 2 (increments: half-point, full point) × 4 (system: GT, T2, GTTS, VITS) mixed factorial ANOVA with repeated measures on system. There was a significant main effect of system, $F_{(3, 300)}$ = 241.54, p < 0.001. Post-hoc t-tests with Holm-Bonferroni correction for multiple comparisons showed that ratings for all

Table 4: *MOS for different measures (collapsed across increments) and scale increments (collapsed across measure) with means for each measure/increment across all systems in the last row. Significant differences (p < .05) are indicated with "<".*

|  | Measure | | | Increment | | |
|---|---|---|---|---|---|---|
|  | Nat. | | Qual | 1 pt | | .5 pt |
| **GT** | 4.16 | = | 4.20 | 4.23 | = | 4.12 |
| **GTTS** | 2.26 | < | 2.88 | 2.44 | < | 2.70 |
| **T2** | 2.95 | = | 3.24 | 3.0 | = | 3.18 |
| **VITS** | 3.46 | < | 3.77 | 3.65 | = | 3.58 |
| | 3.21 | < | 3.53 | 3.30 | = | 3.40 |

systems significantly differed from one another across all versions of the test, p < 0.001. The main effect of measure was also significant, $F$ (1, 100) = 12.58, p < 0.001. Overall MOS was higher for quality ratings (M=3.52, SD=0.80) than for naturalness ratings (M=3.21, SD=0.90).

There were significant interactions between system and measure, $F$ (3, 300) = 7.0, p < 0.001, and between system and increment, $F$ (3, 300) = 4.24, p < 0.01. The simple main effect of measure was significant only for VITS and GTTS, while the simple main effect of increment was significant only for VITS. In other words, ratings of these systems were most impacted by differences in the scale and instructions. MOS for quality and naturalness ratings (collapsed across increment) and for half- and full-point scales (collapsed across measure) are shown in Table 4. Neither the main effect of increment nor any other interactions were significant at p < 0.05.

# 5. Discussion

## 5.1. Survey of Interspeech and SSW papers

Our survey showed both variability and underspecification in how MOS is used in TTS evaluation. In particular, there was variation in what MOS was said to measure, and very little detail was provided regarding rating scale labels and increments or instructions to participants. The lack of consistency among those who did report these aspects of their method makes it hard to conclude that those who left out the details are all measuring MOS in the same way.

This would not be so concerning if these details were completely inconsequential. Our analysis of the surveyed papers, however, showed that ratings of ground-truth audio varied depending on whether the researchers said they were measuring naturalness or quality, with higher MOS reported for quality ratings. The difference we observed between MOS for half-point and full-point increment scales was not significant, however, the small number of data points made this result inconclusive. Furthermore, the amount of variation in listening tests carried out under very different conditions, and the overall lack of details about the methods, made it difficult to draw firm conclusions from this analysis. For this reason, we followed up with an experimental evaluation of how differences in listening test methods might influence MOS.

## 5.2. Experimental evaluation of differences in MOS methodology

The results of the experiments described in section 4 showed that while the scale increment did not affect the ratings of most systems, it did boost MOS for the system with the lowest ratings. Instructing participants to rate naturalness, meanwhile, resulted in lower MOS on average than asking them to rate overall quality, even with no changes to the scale labels. This was in line with our analysis of MOS in the papers we surveyed. However, this difference was driven largely by higher quality ratings for two of the systems, GTTS and VITS.

On the one hand, it is important to note that differences in MOS between different systems were far larger than differences due to instructions or scale increment. The *ranking* of the systems was consistent even if MOS varied in absolute terms. Nonetheless, the results suggest that caution is merited when it comes to using, reporting, and interpreting MOS. The approaches we evaluated received far lower MOS, in absolute terms, than in the original evaluations of these approaches [15, 16, 14]. These low numbers may elicit skepticism, but the

ratings of the GT audio were not out of the ordinary (4.18 overall), and the TTS stimuli were all synthesized with pre-trained models using the official and publicly available implementations. Our results are also consistent with the findings of [20], who obtained lower MOS when they re-evaluated previously evaluated systems.

It was not the aim of the paper to show that any of the systems we surveyed were good or bad, but this illustrates the variability and potential unreliability of MOS. The low ratings may have been a consequence of different testing conditions or different listener characteristics, but another possibility is that the relative quality of the systems included in the evaluation played a role. As shown by [21], listeners tend to make use of the full range of scoring options available to them. This means that differences between ratings of TTS systems are amplified when the quality of all of the systems being compared is high. Differences in testing methodology could potentially heighten this effect even further, which could become increasingly problematic as the overall quality of TTS systems improves.

We are not the first to call attention to these issues. The pitfalls of MOS and widespread underreporting of listening test methodology have been pointed out before [1, 2, 6]. Yet our findings, and those of [7], make it apparent that these problems have not gone away. Suggestions that we expand MOS evaluations to make them more well-rounded and robust [22, 23] or even move away from MOS altogether [1] have not been widely adopted, as evidenced by the surveyed papers overwhelmingly using one or two numbers to answer the question, "how good is my TTS system?" So why do these issues persist?

One reason may be a lack of awareness about the TTS evaluation literature in large swaths of the TTS community. Many researchers may still assume that others are doing their evaluations in the same way and may not think these choices matter much. Impactful work on TTS is also published widely in non-speech technology conferences, such as prominent machine-learning conferences, where there may be even less discussion around these issues. Hence, continuing to look critically at evaluation methods, empirically testing the effects of different methodology, and pushing for more rigorous evaluation standards, are all important goals which should reach a wider audience.

Beyond this, however, concrete steps are needed to encourage and facilitate greater methodological rigor and better reporting. Page limits mean that researchers must make choices about which details to include. If the peer review process does not stress the importance of evaluation methods, authors are unlikely to make this a priority.

It is outside of the scope of this paper to suggest alternatives to MOS, but we do suggest a number of concrete steps that could be taken to improve transparency and consistency in carrying out and reporting TTS evaluations.

## 5.3. Concrete suggestions for researchers, conference organizers, and reviewers

One possible area for improvement is the guidelines provided by conference organizers and followed by reviewers. Setting clear expectations about which details of TTS evaluations need to be reported would help ensure that authors attend to these details and bear them in mind when writing papers. Hence, our first three suggestions are as follows:

- Conference and workshop organizers should lay out minimum requirements for reporting evaluation methods, such as scale increments, labels and a brief description of the most

crucial instructions to participants, and include these guidelines in the paper template and in instructions to reviewers.

- Reviewers should be attentive to the absence of these details and suggest that the authors include them.
- Peer review processes, article length limits, etc., should be set up so that accurate reporting of experiments is incentivized over, say, trying to fit in additional bells and whistles.

In addition to the steps above, it would be helpful to maintain shared tools and platforms for TTS evaluation. This has been done by the organizers of the Blizzard Challenge, a speech synthesis challenge which has been run annually since 2005 (with the exception of 2022) [17, 18, 19, 24, 25]. This challenge requires participants to use the same evaluation tasks for all submissions, and has kept these tasks similar over the years. The website of the Centre for Speech Technology Research at the University of Edinburgh [6] provides samples and data from previous challenges going back to 2008, as well as scripts for running listening tests. These kinds of resources can be directly used by researchers both to concretely grasp how others are evaluating their TTS systems and to run similar evaluations. So in addition to clearer guidelines for authors and reviewers, we suggest the following:

- Researchers should consider making the code for their listening tests publicly available, both to help readers understand how the evaluation was carried out and to contribute to a common set of evaluation tools.
- The external demo pages typically used to share examples of stimuli could also include images of the evaluation interface and examples of the questions and scales.
- Conference organizers should consider making toolkits such as the ones used in the Blizzard Challenge available to authors and encouraging their use, where applicable.

Finally, our results have implications for how MOS should be interpreted. The fact that our listening tests resulted in surprisingly low MOS values for TTS systems using pre-trained models and official, public repositories (while scores for natural speech were not out of the ordinary) leads us to a final recommendation:

- Researchers, and reviewers, should take absolute MOS values with a grain of salt, and should not directly compare scores obtained by different systems under different evaluation conditions or hold expectations about what constitutes an "acceptable" MOS value for a given system. In particular, a baseline method receiving a lower MOS value compared to what the same approach received in other, earlier publications, does not imply that the baseline is flawed.

## 6. Conclusions

We surveyed a number of recent Interspeech and SSW papers on speech synthesis and determined that many authors under-report or are inconsistent about how they carry out MOS tests, especially with regards to the rating scales they use, instructions to the participants, and what they aimed to measure with MOS. We found that these choices may have impacted MOS in the surveyed work, and followed up with an empirical evaluation of the effect of scale increments and instructions. We found that these choices were impactful, at least for the ratings of some systems. Based on our findings we formulated a number of recommendations that could help researchers be more clear and

consistent in how they carry out and report evaluations. Future work could look into whether systems that are more similar in terms of quality and naturalness are more strongly impacted by methodology, or explore aspects we were not able to assess here, such as the effect of variations in scale labels. We also join other researchers in suggesting a more critical stance towards the use of single-item MOS evaluations and an exploration of how we can most meaningfully evaluate TTS going forward.

## 7. Acknowledgements

This work was partially supported by the Wallenberg AI, Autonomous Systems and Software Program (WASP) funded by the Knut and Alice Wallenberg Foundation and by the Swedish Research Council project Connected (VR-2019-05003).

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
