# OpenReview forum: "Stuck in the MOS pit: A critical analysis of MOS test methodology in TTS evaluation"
_Interspeech.org/2023/Workshop/SSW — SSW12_

### Official Review · Reviewer_1Gzm · 2023-05-17
**A brief analysis of reported MOS scores in Interspeech and SSW**

**Rating:** 7
**Confidence:** 5

**Review:**

The paper introduces an analysis over reported MOS scores for naturalness/quality of synthesised speech across several papers published in the 2021 and 2022 Interspeech and 2021 SSW conferences. The analysis looks into understanding if the different setups (i.e. questions asked and MOS scale divisions) influence the overall ratings of the samples. The authors then perform an in-house listening test to mimic these two listening test axes.

The subjective evaluation for speech synthesis is indeed of great importance and poses numerous problems in direct comparisons between the different reports. Automatic MOS evaluations would solve this issue, but a commonly agreed method is not yet available. The results of this study are of interest to the wide community of SSW attendees.

Key strengths:
- an extensive analysis over MOS reported results across 133 papers
- some initial conclusions over the inaccuracies of the reported MOS results are drawn
- a large listening test is conducted

Weaknesses
- reporting the ground truth MOS scores can be seen as a reference point across listening tests, especially if the papers also include samples from other baseline systems.
- no definite conclusions/potential improvements are drawn/proposed -- all the disadvantages of using MOS scores are known in the community
- the results of the authors' listening test measured with t-tests, do not seem as problematic as the authors claim -- I do not believe that a definite conclusion on the design of a listening test can be drawn from this simple setup.
- more similar TTS systems could have been included in the test, such that only minor differences would have been expected between the systems
- I could not find any mention if the ground truth samples are vocoded or natural ones.

Potential developments:
- release the MOS scores of the listening test and samples for better automatic MOS predictions
- create a unified platform for listening test easily adoptable by researchers (a simpler alternative to webMushra)
- add more dimensions to the listening test
- add more analysis with respect to the listener's profile (gender, age, nationality)

The paper is clearly written and references seem adequate.

---

### Official Review · Reviewer_5DuA · 2023-06-01
**The paper follows on the work of Wester et al. to analyse the robustness of reported conclusions based on MOS - I think the work should be accepted despite lacking a more thorough bibliographic analysis**

**Rating:** 8
**Confidence:** 5

**Review:**

***** Key Strength of the paper

- meta-analysis + additional subjective evaluation to determine the biases due to the use of the ACR

***** Main Weakness of the paper

- references missing
- concrete suggestions underwhelming

***** Novelty/Originality, taking into account the relevance of the work for the SSW audience

The work follows a series of previous studies questioning the robustness of results presented using MOS. It is really relevant for SSW

***** Technical Correctness

The work looks technically solid to me. I would have liked a bit more justification about the choice of the statistical tests used and about the posthoc test.

***** Quality of References

It is undeniable that the justification of the work is well presented using the literature. Nonetheless, the submission misses key works about the MOS test and TTS evaluation. This includes This includes:
  - Clark, Robert AJ, et al. "Statistical analysis of the Blizzard Challenge 2007 listening test results." Proc. BLZ3-2007 (in Proc. SSW6) (2007).

    which provides guidelines which should be discussed in the current paper

  - Le Maguer, Sébastien, Simon King, and Naomi Harte. "Back to the Future: Extending the Blizzard Challenge 2013." Proc. Interspeech. 2022.

    which also provides an analysis of MOS (it is actually unclear if it is part of the paper used for the analysis or not)

  - ITU-T. "P. 85. a method for subjective performance assessment of the quality of speech voice output devices." International Telecommunication Union, Geneva (1994).

  - Hinterleitner, Florian, et al. "An evaluation protocol for the subjective assessment of text-to-speech in audiobook reading tasks." Proceedings of Blizzard Challenge (2011).

    the last two references provide an alternative to the P.800 with more dimensions to be evaluated. These references would be more relevant for the discussion section than the [16].

***** Clarity of Presentation

The paper is clear and easy to follow.

***** Suggestions for improvement

- the authors should make it more clear that they are updating the work of Wester et al. 2015.
- it is unclear if the authors checked that the analyzed papers not explicitly state that the evaluation actually referenced the P.800. If that is the case, it can be argued that the submission actually states the dimension they are evaluating.
- table 4 looks not finished
- why is the sentence "concrete suggestions for..." in bold?
- the concrete suggestions are too vague:
  + the blizzard already provides a standard; conferences could enforce it
  + while the demonstration page would be an improvement, it still lacks some key pieces of information which can only be solved by providing recipes. Standard toolkits already exist and, in the same way that open datasets should be reinforced, providing a recipe for one of these toolkits could also be enforced.

---

### Decision · Program_Chairs · 2023-06-14

**Decision:**

Accept

**Comment:**

SSW2003 received 45 papers. The acceptance rate is 82%. We are pleased to inform you that your paper has been accepted by the SSW2023 Program Committee. Please read the reviews carefully and submit your camera-ready paper by June 28th. Most reviewers performed a detailed review. Please answer to their questions and consider their comments. Note that camera-ready papers are credited with one extra page to allow authors to consider reviewers’ suggestions. So max 7 pages in total including figures & refs.
The deadline for submitting the revised version (with full non-anonymized authors and refs!) is 28th June.